# The Design and Ground Test Verification of an Energy-Efficient Wireless System for the Fatigue Monitoring of Wind Turbine Blades Based on Bistable Piezoelectric Energy Harvesting

**DOI:** 10.3390/s24082480

**Published:** 2024-04-12

**Authors:** Theofanis Plagianakos, Nikolaos Chrysochoidis, Georgios Bolanakis, Nikolaos Leventakis, Nikolaos Margelis, Manolis Sotiropoulos, Fotis Giannopoulos, Grigoris-Christos Kardarakos, Christos Spandonidis, Evangelos Papadopoulos, Dimitris Saravanos

**Affiliations:** 1Control Systems Lab, School of Mechanical Engineering, National Technical University of Athens, 15780 Athens, Greece; gbolanakis@hotmail.com (G.B.); nikoslev@gmail.com (N.L.); nmargelis@gmail.com (N.M.); egpapado@central.ntua.gr (E.P.); 2Department of Mechanical Engineering and Aeronautics, University of Patras, 26504 Patras, Greece; nchr@mech.upatras.gr (N.C.); gregorykard@yahoo.gr (G.-C.K.); saravanos@mech.upatras.gr (D.S.); 3PRISMA Electronics, 17564 Paleo Faliro, Greece; manolis.sotiropoulos@prismael.com (M.S.); fotis.giannopoulos@prismael.com (F.G.); c.spandonidis@prismael.com (C.S.)

**Keywords:** piezoelectric, energy harvesting, wind turbine blade, bistability, demonstrator, ground test

## Abstract

A wireless monitoring system based on piezoelectric energy harvesting (PEH) is presented to provide fatigue data of wind turbine blades in operation. The system comprises three subsystems, each respectively providing the following functions: (i) the conversion of mechanical to electric energy by exploiting the bistable vibration of a composite beam with piezoelectric patches in post-buckling, (ii) harvesting the converted energy by means of a modified, commercial, off-the-shelf (COTS) circuit to feed a LiPo battery and (iii) the battery-powered acquisition and wireless transmission of sensory signals to the cloud to be elaborated upon by the end-user. The system was verified with ground tests under representative operation conditions, which demonstrated the fulfillment of the design requirements. The measurements indicated that the system provided 23% of the required power for fully autonomous operation when subjected to white noise base excitation of 1 g acceleration in the range of 1–20 Hz.

## 1. Introduction

Monitoring the aeroelastic vibration response of wind turbine blades drives their reliability and is essential for maintenance intervals’ scheduling [1]. In situ access to the blade requires special equipment and experienced personnel, especially in the case of off-shore wind energy parks, and therefore leads to a high cost for the inspection and replacement of components of the monitoring system, such as batteries. Self-powered monitoring systems capable of harvesting energy from physical sources offer a viable solution to this problem. Among the available energy-harvesting technologies, those based on electromechanical energy conversion during vibration by means of piezoelectric transducers offer specific advantages related to power density and ease of implementation [2]. Various Piezoelectric Energy Harvesters (PEHs) have been developed in the last 25 years, mainly as demonstrators, in a wide range of applications [3,4,5,6,7,8]. In the last 15 years, nonlinear multi-stable PEHs have gained more attention due to their superiority in power output over a wide range of excitation frequencies [9,10,11,12], which is the case in most operational environments. 

Nonlinear energy harvesting technologies aim to develop power-autonomous devices in various sectors, such as aeronautics [13,14], infrastructure [15], medicine [16] and wearables [17], as parts of IoT networks. The electromechanical concept of multi-stable vibration may be based on two major mechanism categories: (i) electromagnetic forces applied via the opposite polarization of magnets located on the vibrating piezoelectric structure and the supporting device [18,19] or (ii) base excitation and axial compressive forces acting on a piezoelectric structure up to post-buckling [20,21]. The PEH developed for the monitoring system presented herein is based on the latter functional principle. Nevertheless, both types of harvesters are included in this literature survey on the basis of a Technology Readiness Level of 4–5, meaning being ground-tested under representative operating conditions.

Many of such bistable PEH demonstrators have been developed to be subjected to wind energy excitation. Alhadidi and Daqaq [22] studied the performance of a bistable wake-galloping flow energy harvester, which harvested 6 mW of power at a wind velocity of 6 m/s. Zhao et al. [23] formulated a distributed coupled aero-electro-mechanical model to investigate the dynamic behavior of a cantilever-type vibro-impact triboelectric energy harvester. A nonlinear response due to impact led to a harvested power of 290 μW at a wind speed of 10 m/s. Li et al. [24] studied the performance of a bistable energy harvester under hybrid base vibration and galloping, which harvested a rising power of 0.16 mW at 18 Hz on a resistance of 200 kOhm for a wind speed of 3.6 m/s. Liu et al. [25] investigated the performance of a bistable piezoelectric wind energy harvester under a uniform airflow using a finite element analysis and tests, achieving 7.2 mW of power at 8 Hz on a resistance of 60 kOhm for a wind speed of 14.5 m/s. Zhao [26] developed a bistable galloping energy harvester for power generation from concurrent wind flows and base vibration, which provided the maximum power at 12 Hz. Pan et al. [27] designed a zigzag-shaped energy harvester based on nonlinear magnetic coupling to harvest energy from a concurrent wind flow and base excitation, and they extracted 10 μW at 14 Hz and a base acceleration of 0.5 g. Li et al. [28] developed a wind energy harvester based on magnetic coupling, which consisted of piezoelectric beams, magnets and coils, and they achieved 0.31 mW from one beam and 2.154 mW from one coil at a wind speed of 7 m/s on a resistive load of 200 kOhm.

In the field of transportation engineering, Ansari and Karami [29] developed a PEH device based on the controlled buckling of a steel sheet with piezoelectric layers or patches, capable of producing tens of mWatts from passing car traffic. Zhang et al. [30] proposed a bistable piezoelectric energy harvester with magnets for harvesting energy from bridge vibrations triggered via moving vehicles. Bistable harvesting devices for medical and biological applications have also been developed, such as the demonstrator developed by Ansari and Karami [31], which is based on a buckled beam in sub-cc scale and provides enough power to operate a pacemaker. Qian et al. [32] developed a bio-inspired bistable energy harvester exploiting fish maneuvering and the surrounding fluid in order to achieve an autonomous power supply for continuous fish tracking. In addition to research demonstrators, several multi-purpose patents describing PEH devices based on a multi-stable response, such as [33,34,35,36,37], have been submitted or filed.

Thus, it can be concluded that no bistable PEH device has been developed so far for application on vibrating wind turbine blades. In the following sections, such a device is presented, and it was used to provide power to data acquisition and wireless transmission hardware in the context of a novel integrated system for fatigue monitoring. Based on the modeling and experimental verification of a PEH presented in [38], the full system is described, and ground tests conducted under operational conditions are outlined, providing evidence of the system’s capabilities in terms of power harvesting and overall performance. The following sections are structured according to the typical design and development process, starting from the requirements set by the end-user, continuing with the description of each subsystem and its integration into the full system, and concluding with experimental verification.

## 2. Design of the Energy-Efficient Monitoring System

The EnAuSy blade monitoring system (Figure 1) comprises three units: (I) an electromechanical energy conversion unit (EMCU), (II) an energy-harvesting circuit unit (EHCU) and (III) a data acquisition and wireless transmission unit (DAWTU). The EMCU and HCU include the PEH device and the corresponding circuit, respectively, and they interact to feed a LiPo battery, which is part of the EHCU, to provide power for the operation of the DAWTU. The DAWTU includes piezopolymer sensors for strain monitoring at selected locations along the blade, and it accomplishes wireless transmission to the cloud by means of a gateway and blade tower infrastructure. Before proceeding to the detailed description of each particular unit, the requirements set via the operator are presented.

### 2.1. Design Requirements

The design requirements of the blade monitoring system stem from the operational environment and safety assessment. The PEH will be installed in the blade and will, thus, be subjected to a low-frequency excitation range up to 15 Hz induced via aerodynamic and inertia loads. Blade stability and structural integrity issues arise due to the local addition of weight, which were considered in the EMCU design phase to achieve a lightweight, low-volume, resilient component for electromechanical conversion. Additional installation requirements are related to the routing of cables and the positioning of small electronic components for data acquisition and wireless transmission. The main requirements set via the operator, along with the specific ones related to the subsystem responsible for the calculation of fatigue monitoring (thus, the DAWTU system), are summarized in Table 1.

### 2.2. Electromechanical Conversion Unit (EMCU): Bistable PEH

The electromechanical subassembly consists of a composite beam, manufactured using hand layup to achieve maximum flatness, which was axially loaded in compression up to the post-buckling regime. Two piezoelectric transducers were placed symmetrically at the mid-span, at the top and at the bottom face (Figure 2). DuraAct P876.A11 [39] transducers were selected, which offer great deformation capabilities while including piezoceramic material and, thus, exceeding piezopolymer sensors in electromechanical conversion. Their terminals are connected to the EHCU.

The composite beam was sized to 368 × 35 × 2.5 mm^3^ in order to be in the post-buckling regime under an axial load in the range of 300–400 N. Sizing was based on previous numerical and experimental studies [38], which indicated that the beam should not be too compliant to achieve the desired power of 1 mW (Table 1).

Figure 3a illustrates the digital mockup of the EMCU connected to the EHCU. Axial compression of the composite piezoelectric beam was achieved by means of a mechanical assembly consisting of a compression mechanism and its supporting steel frame. The mechanism includes a μm-resolution thread screw and sheet metal clamps assembled with screws in tight tolerance with the frame. The frame was designed by means of finite element analysis to safely endure the compressive reaction loads with acceptable deformation (Figure 3b). It was manufactured using laser cutting and sheet metal forming. The design strategy encompasses the tuning of the concentrated mass placed symmetrically at the center of the beam to achieve the desired power output and weight requirement. Each mass part weighs approximately 72 gr. By placing two mass parts, a weight of 1134 gr was achieved, as shown in Figure 3c.

The compression mechanism, thanks to the μm-resolution thread screw, allows for the tuning of the compressive load applied to the composite beam. This attribute differentiates this harvesting unit from other configurations containing piezoelectric transducers since it can reliably and accurately set the composite beam in the post-buckling regime while shifting its eigenfrequency so that it matches the expected range of the excitation frequency of the blades. By means of this tuning process, the maximum output power from the transducers is achieved.

From a design point of view, using a larger piezoelectric patch would increase the harvested power. However, it would also lead to higher axial loads required to achieve post-buckling and, thus, to a heavier supporting frame, as well as a higher cost due to the customization of the piezoelectric patch and compression screw.

### 2.3. Energy-Harvesting Circuit Unit (EHCU)

The EHCU is based on the SPV1050, which is an off-the-shelf, low-power, high-efficiency energy harvester and battery charger implementing the maximum power point tracking function (MPPT) and integrating the switching elements of a buck-boost converter. Circuit customization for the blade monitoring application includes the parametrization of the harvesting board, as well as design optimization and the in-house implementation of full-wave rectifiers with voltage overshoot protection [40]. The circuit selection for the nonlinear system described herein was presented in [41].

A schematic representation of the EHCU architecture is shown in Figure 4a. The unit integrates a single SPV1050-based harvesting board (Figure 4b) from STMicroelectronics (Geneva, Switzerland) in order to minimize power losses attributable to the SPV1050 self-powering requirements. Two piezoelectric transducers are connected to the harvesting board via dedicated full-wave rectifiers (Figure 4c). The output stage of the rectifiers was parallelized, adding the electric power contributions of each individual piezoelectric transducer and distributing them to the harvesting board. Moreover, a BZX85C18-TR Zener diode with a breakdown voltage of V_z_ = 18 V was added to protect the harvesting board from overvoltage. In the final stage, a LiPO battery was connected to the harvesting board’s output. The harvesting board was parametrically tuned by setting the values of the electronic components according to the recommendations in the manufacturer’s datasheet while taking into account the electrical characteristics of the piezoelectric transducer output (Table 2).

The full-wave rectifier circuit was simulated using the LTspice XVII (v17.0.27.0) by Analog Devices Inc. (Wilmington, MA, USA) software in order to select the most appropriate diode for the studied blade monitoring application. The piezoelectric transducer was modeled as a sinusoidal current source with an amplitude of 70 uA and a frequency of 5 Hz. A schematic representation of the simulated system is shown in Figure 5. The output (harvested) mean power was measured for a varied load of 400 kΩ, 500 kΩ, 600 kΩ and 700 kΩ. The efficiency of two widely used diodes, 1N4148 and CUS520, was investigated and compared against the efficiency of an ideal diode. The results presented in Figure 6 demonstrate that both diodes are by far inferior in comparison with an ideal diode. Concerning the commercially available diodes, the circuit based on the 1N4148 diode is the most effective for the particular application, exhibiting a maximum output power of 260.38 uW and an efficiency of 92.6% in the case of a 600 kΩ load.

### 2.4. Data Acquisition and Wireless Transmission Unit (DAWTU)

The DAWTU subsystem comprises two physical devices, the collector and the gateway. Its role is to acquire measurements from piezopolymer sensors for strain monitoring and transmit this data to the remote user for additional analysis and visual representation via a web application, which was also developed in-house. The design of the DAWTU subsystem, both its hardware and its embedded software, is based on the PrismaSense™ technology [42], which was upgraded to serve the aforementioned functionalities. The collector is powered from the PEH subsystem via a battery; it is placed inside the wind turbine’s blade, and it interfaces with two types of piezopolymer sensors: 2× DT1-028K [43] for measuring the strain and 1× MiniSense 100 [44] for the acceleration. Thus, it samples data crucial to the blade’s fatigue under a predefined rate and wirelessly transmits the information to the gateway. The latter, which is placed inside the nacelle of the wind turbine and is connected to the power and network structure of the wind energy park, is responsible for the reception of the sampled data sent via the collector and its proper forwarding to the cloud. A web application, the presentation of which exceeds the scope of the present manuscript, takes over the role of information processing and visualization in a user-friendly way. The architecture of the system is displayed in Figure 7.

A deeper understanding of the DAWTU subsystem can be obtained by delving into the architecture and the modules comprising the devices that the DAWTU subsystem consists of. The collector device constitutes three sub-modules, namely the following: (a) the Signal Conditioning Unit (SCU), which amplifies, properly filters and converts the analog signal (under a constant frequency) into a digital one—thus, it is responsible for interfacing with the sensors; (b) the Control Unit (CU) which is responsible for configuring the functionalities of the collector, setting the sampling frequency (implemented via the SCU), activating the WCU when the memory usage has reached a critical level, safely transmitting all necessary information to the WCU, restoring the sampling functionality, monitoring the battery level and deactivating the WCU when the wireless transmission of the data from the WCU is completed in order to minimize the power consumption; (c) the Wireless Communication module (WCU) that is charged with the proper formatting of the data and their wireless transmission via a Bluetooth Low Energy (BLE) protocol. In Figure 8, the architecture and the data flow between the collector’s submodules are depicted in a block diagram format. By employing this architecture, data from the piezoelectric sensors are collected and transmitted via the collector in two distinct phases: a data collection phase, during which data from the piezoelectric sensors is acquired at the Control Unit via the Signal Conditioning Unit and are temporarily stored before being transmitted to the gateway. This process lasts approximately 146 s in order to acquire data from three sensors at a sampling rate of 100 Hz. Consequently, the transmission of the collected data to the DAWTU’s gateway is accomplished within 20 s at most. After the data transmission has been accomplished, the data acquisition phase starts again. Therefore, the DAWTU does not operate in real time.

The block diagram of the gateway architecture is shown in Figure 9.

The gateway remains in a standby status, awaiting data. When such data are received and properly formatted (into proper JSON files, including all necessary data—measurements, ID, timestamp, etc.), it transmits all data via Ethernet and over an MQTT protocol to the cloud, from which the web application can retrieve them, with a user-friendly GUI, for further processing and visual representation.

In particular, the web application offers the ability to retrieve, download and process data. The sensory data are processed via the Rainflow Counter Method algorithm [45] to determine the load cycles of the blade. The algorithm is applied to the range of the raw data selected by the remote user. The resulting cycle counts are then displayed in histogram format. Thus, the elaboration of the raw data in the EnAuSy web application enables the estimation of the remaining useful life of the blade.

The collector and the gateway devices are properly encased in order to meet the specific environmental conditions of the blade and the nacelle, in which these devices are planned to be placed. The casings were selected to ensure the compliance of the collector and the gateway with the geometrical limitations, ease of access and functional terms while also fulfilling all stakeholders’ requirements. In Figure 10, there are photos of the physical devices in their final enclosures.

## 3. Integrated Blade Monitoring System

The Integrated Blade Monitoring System comprises the three main subsystems described above, which provide electromechanical conversion, energy harvesting, data acquisition and wireless transmission. In the current application, there are three EMCUs connected to corresponding EHCUs placed in a harvesting circuit subassembly, which provides power to the DAWTU (Figure 11).

The harvesting circuit sub-assembly integrates three sets of dual full-wave rectifiers (Figure 4c) along with a harvesting board (Figure 4b). Each set is responsible for capturing energy that is produced via a pair of DuraAct P876.A11 transducers and saving it to an incorporated custom battery pack. Six single-cell lithium-polymer (LiPO) batteries in a parallel configuration are used. The nominal terminal voltage of the battery pack is 3.7 V, reaching a total nominal capacity of 5700 mAh. The housing was designed and developed in-house in order to facilitate assembly operations and protect the electrical components from dust and other environmental hazards (Figure 12). The harvesting circuit subassembly ensures the constant operation of the entire system for a minimum of 30 days; the harvested energy gain is not considered. The powering of the DAWTU subsystem is realized through a 2-pin circular connector. The assembled subsystem, excluding the top cover, is presented in Figure 13.

The Integrated Blade Monitoring System can support many types of wired sensors that provide signals useful for determining the fatigue loading of the wind turbine blade, such as piezoceramic sensors and piezopolymer accelerometers. The signals from these sensors are received via the DAWTU. As part of the DAWTU, the collector is responsible for receiving the data from the wired sensors and then transmitting them to the gateway through Bluetooth Low Energy (BLE). The gateway is connected to the wind turbine’s network infrastructure so that the accumulated data can be saved on a remote server.

The placement of all the subsystems is displayed in Figure 14. All subsystems except the gateway are fixed inside the blade section to ensure stability during vibration. The EMCUs must be placed away from the root of the blade so that the amplitude of the blade’s oscillation is greater, resulting in larger amounts of energy being harvested. It is also important that they are placed vertically to the length axis of the blade and the plane of the blade’s rotation to harvest the energy from the fundamental bending mode of the blade. In Figure 14a,b, the wind turbine blade installation of an EnAuSy configuration with two EMCUs is shown. The harvesting circuit sub-assembly and the collector of the DAWTU subsystem are placed in close proximity to the EMCUs so that the energy losses in cabling are minimized. The measuring sensors can be installed at different locations to maximize their effectiveness, depending on their type. The gateway is installed in the nacelle of the wind turbine so that it can be connected to the turbine’s network infrastructure while still being able to receive the data from the collector through BLE.

## 4. Ground Tests

The ground tests were performed in two subsequent stages. First, the harvesting subsystem (EMCUs and the EHC subassembly) was tested to validate performance in terms of harvested power. In parallel, testing of the Data Acquisition and Wireless Transmission Unit (DAWTU) was conducted by independently exciting a compliant, pure-epoxy beam instrumented with piezopolymer sensors (DT1-028K) [43]. The signals from these sensors were acquired and transmitted to the cloud. In a second stage, the integrated system, including the harvesting subsystem and DAWTU, was tested under typical operating conditions. 

The lab equipment required during the testing campaign was based on the concept of the energy harvesting experimental studies described in [46]. Based on this concept, the testing frame consists of an aluminum horizontal platform laterally oscillated on a pair of horizontal slides. Lateral oscillation was achieved via an electromagnetic shaker with a loading capacity of up to 196 N that was mounted on the platform with the intervein of a dynamic load cell. Simultaneously, the acceleration generated was measured with uniaxial accelerometers bonded onto each specimen. The actuation and DAQ were performed via an NI DAQ chassis providing a 2.5 kS/sec input and output rate. Finally, when it was required, lateral specimen displacement was acquired via a non-contact laser displacement sensor. 

### 4.1. Harvesting Subsystem

For the particular application of monitoring the wind turbine blade, the harvesting subsystem demonstrator includes three independent EMCUs, as shown in Figure 15a. Independence here is related to the design choice of each pair of DuraAct transducers connected to its own dual-channel full-wave rectifier as part of the harvesting circuit subassembly. Previous simulations and tests on similar composite piezoelectric beams [38] quantified the required prestress to get the composite beam deep in post-buckling to harvest the maximum vibration energy. Due to the absence of a static load cell capable of measuring the axially generated load on the specimen, the only way to identify the ideal prestress level on each specimen remained the first bending modal frequency variation as a function of the resulted lateral specimen displacement due to the applied axial load via the micrometer screw. The pre-loading procedure was performed independently for each of the EMCUs that was mounted on the aluminum platform, which laterally oscillated via the electromechanical shaker with white noise excitation in the range of (0–200) Hz with simultaneous acquisition of the specimen acceleration. Using the measured acceleration and the applied dynamic load from the shaker, the FRFs were extracted. Loading towards buckling was performed stepwise, based on the lateral displacement generated at the specimen center as the axially applied displacement slowly increased. This procedure, which may be, thus, characterized as a “tuning” procedure of the EMCU, serving the identification of the most promising prestress levels for highest power extraction, is illustrated for one of the EMCUs in Figure 15b. Additionally, Figure 15c demonstrates a simulated phase plot of the beam response, aiming to highlight the specimen’s bistable behavior and nonlinear vibration at the post-buckling regime. The simulation refers to 8 Hz tonal excitation and a 60μm axial compressive displacement, resulting in a 13.1 Hz first bending modal frequency on the specimen. The ideal prestress “tuning” process led to fundamental eigenfrequencies of 11.0 Hz, 15.47 Hz and 17 Hz for each EMCU beam specimen, respectively, depending on the precise individual boundary conditions and tolerances of each unit. 

The next step of the harvesting system ground testing was the placement of each “tuned” EMCU on a piece of wood, mounted on the aluminum platform, as presented in Figure 15a. The platform was laterally oscillated via the electromechanical shaker, resulting in simultaneous bistable oscillations for all three EMCUs. To provide realistic oscillatory conditions, the actuation signal was white noise in the frequency range of (0–20) Hz. Generated voltages from the six DuraAct piezoelectrics were continuously collected from the harvesting circuit subassembly.

Given the characteristic curve of the battery, i.e., the terminal voltage as a function of the discharge capacity, mean electric power estimations could be carried out based solely on terminal voltage measurements. For the ground tests, the battery pack of the harvesting circuit subassembly was replaced with a single-cell, low-capacity LiPO battery in order to accelerate the power measurements. The battery used is shown in Figure 16a, and it is characterized by a nominal terminal voltage of 3.7 V and a nominal capacity of 250 mAh. To determine the characteristic curve of the battery, a phenomenological model was developed for the operation of the battery under a known constant load, R_L_, as shown in Figure 16b. The battery was modeled as a constant voltage source, V_B_, in series with an internal resistor, R_IN_. The resultant characteristic curve is presented in Figure 16c.

### 4.2. Data Acquisition and Wireless Transmission Subsystem

The experimental setup for the DAWTU subsystem is visually presented in Figure 17 and schematically represented in Figure 18. The test setup included the tonal excitation of a cantilever epoxy test beam via a second electromechanical shaker, as this testing campaign was performed simultaneously with the harvesting subsystem ground tests. The pure resin specimen (1 m long) was selected for this application, as it was very compliant, providing a high oscillation stroke in low frequency ranges, resulting in a high voltage level on the piezopolymer sensors. Two pairs of piezoelectric sensors were placed on each side of the beam. Data from one or two piezopolymer sensors were acquired via the collector and sent to the remote server via the gateway. The sampled data from the collector and the data sent to the server via the gateway were recorded on a computer. The logged data from each device were then compared to each other and to the data from the sensors for verification purposes. Finally, the data acquired from the integrated system (piezopolymer or battery-related) were monitored via a web application. In particular, the raw and/or elaborated data are offered to the remote user in a visual way, as can be seen in Figure 19.

Experiments on the variable excitation frequency, within low frequencies of up to 15 Hz, and/or the amplitude were conducted multiple times in order to verify both the integrity and reliability of the measurements. Data collected via the DAWTU were compared to measurements from an oscilloscope that constantly monitored the sensors’ outputs so that the DAWTU’s data integrity could be verified, i.e., to confirm whether the DAWTU provides correct information regarding the strain/loading of the test specimen. The comparison showed that the DAWTU subsystem responds as expected to the changes in oscillation amplitude and frequency. Figure 20 presents the response signal to the oscillation of a nominal frequency of 8 Hz, as recorded via the collector, for two different amplitude values (1 V and 2 V) of the generator. Figure 21 is an indicative instance of Figure 20. Both figures show that the increase (in this case, the doubling) of the amplitude of the shaker leads to an actual increase (doubling) of the amplitude recorded via the collector. Figure 22 displays the signal and the Fourier transformation result from the oscillation of the elastic test beam under two indicative frequency values (5 Hz and 12 Hz, respectively). The figure demonstrates that the recorded frequency is the same as the shaker’s oscillation frequency and that the changes in the shaker frequency can be monitored via the collector.

Apart from those tests, which mostly demonstrate the performance quality of the DAWTU, tests were also performed to record the power consumption of the subsystem since it is the one that is powered via the PEH, and thus, little consumption is desired. A typical instance of the power profile during the different phases of operation (e.g., sampling, wireless transmission, etc.) in one full cycle of operation (from the initiation of operation until the first wireless transmission of all collected data to the gateway is completed) is presented in Figure 23. The power consumption of the collector presented in this figure corresponds to the power drawn from the collector in order to acquire data from three sensors for a sampling duration of 146 s at a sampling rate of 100 Hz and to transmit the collected data to the DAWTU’s gateway, which is accomplished within 20 s at most. The data transmission process is the most demanding phase in terms of power, as can be seen in Figure 23. The mean value of power required for the entire process under typical operating conditions was estimated to be 32 mWatt. As soon as the wireless transfer of data is complete, the wireless data-transferring unit is deactivated, and the data acquisition process restarts.

### 4.3. Integrated System

The integrated system is shown in Figure 15a. It was tested under white noise excitation in the range of [0–20] Hz and a 1 g base acceleration, which is considered representative of a typical operational state near the mid-span of a 55 m blade at windspeeds of 10 m/s. Based on the derived battery characteristic curve, mean harvested power estimations were carried out by measuring the battery’s terminal voltage in two independent tests, each one lasting for a duration of approximately 2.5 h: (i) a standalone battery autonomously feeding the DAWTU and (ii) a battery feeding the DAWTU while being continuously supplemented with the electromechanical subsystem. A comparison of the two tests indicates the reduction in the total power consumed, which actually is the harvested power contributing to the operation of the DAWTU. In Figure 24 and Figure 25, the battery terminals’ voltage is respectively shown for the two independent experiments.

Based on these measurements and the characteristic curve of the battery (Figure 16c), mean power consumption estimations were performed for the two independent experiments, as shown in Table 3. It should be noted that the standalone power consumption refers to the maximum power in the transmission unit to ensure a data transfer under the worst circumstances.

From Table 3, it may be concluded that the power provided via the harvesting subsystem demonstrator is 7.18 mW, leading to an energy profit of 13.52%. Considering that the DAWTU requires a mean power of 32 mW under typical operating conditions to provide 146 s of 16-bit data sampled at 100 Hz every 166 s (Section 4.2), a total of 15 EMCUs would be required to fully capture the mean power demand of the DAWTU in the case of a base excitation acceleration of 1 g at a white noise frequency spectrum in the range of 1–20 Hz. In that case, the harvesting subsystem (EMCUs and harvesting subassemblies) could be placed symmetrically in pairs at adjacent blade sections. 

The system’s performance may be further improved by reducing the sampling rate or duration, as well as by placing the harvesting subsystem near the blade tip, where base accelerations are the highest. However, this approach is limited due to the least amount of data required for the operator to accurately estimate fatigue loading cycles and by the memory of the selected Signal Conditioning Unit (SCU). Thus, autonomous operation would require a compromise between the number of EMCUs, the SCU design and the DAWTU duty cycle under the condition of 1 g of mean acceleration excitation. If the power consumption of the DAWTU system is disregarded, and considering a mean power of 7.182 mW supplied from the PEH unit, approximately 122 days would be required to fully charge the battery pack.

## 5. Summary and Conclusions

EnAuSy is an integratedpower-semi-autonomous wireless system for the fatigue monitoring of wind turbine blades. Its power autonomy is based on piezoelectric energy harvesting from the bistable vibration of composite beam specimens in post-buckling and optimized electric circuitry. Ground tests were conducted for the verification of all subsystems’ performance, namely EMCU, EHCU and DAWTU. Each subsystem demonstrated reliability, robustness and functional efficiency, as required. In particular, the pipeline of tasks—from data sampling of a specific frequency to data accumulation in a remote server for storage and visualization purposes—exhibited the required efficacy and effectiveness. The performance of the integrated system was demonstrated under operating conditions of white noise signal excitation in the range of 1–20 Hz at a base acceleration of 1 g.

In the tested configuration comprising three EMCUs, 23% of the required power was provided for a duty cycle of 50% in sampling 13-bit data at 100 Hz, assuming that the base acceleration of the blade vibration was 1 g. Under these conditions, self-powered operation can be supported for approximately 36 days. However, the system design can be adapted according to a wind turbine park operator’s requirements to support fully autonomous operation by considering an appropriate sampling/transmitting frequency, the amount of EMCUs and the installation topology to achieve higher acceleration values. In that case, the system lifespan would be exclusively dependent on the preservation of compressive prestress in the EMCU. Such high-cycle fatigue tests have yet to be performed. 

The tested demonstrator weighs approximately 4.5 kg. It is small and portable, practically fulfilling all the design requirements of Table 1. Compared to battery replacement, the proposed system offers a significant maintenance cost reduction in the long term, mainly due to the minimal downtime of the wind turbine and the cost related to the training and working time of specialized technical personnel who will perform the battery replacement.

The autonomy of the system in terms of power is of potential viability. The compliance of power demands of the DAWT subsystem with the power harvesting capabilities of the harvesting subsystem depends on the size of the blade, the position along the length of the blade, the data sampling rate and the duration. Thus, the EnAuSy system can be tailored according to its application. The on-field implementation of the system is expected to provide valuable evidence of the system’s performance, the effect of environmental conditions and the system’s durability.

## 6. Patents

Self-powered Data Acquisition and Wireless Transmission System of Wind Turbine Blade Fatigue Data based on Energy Harvesting from Piezoelectric Transducers. Submitted to Hellenic Industrial Property Organization on 16 January 2024. 

## Figures and Tables

**Figure 1 sensors-24-02480-f001:**
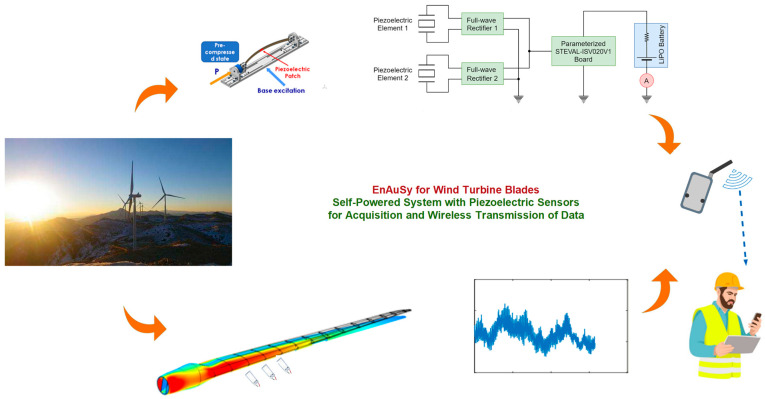
EnAuSy blade monitoring system (see also Appendix A).

**Figure 2 sensors-24-02480-f002:**
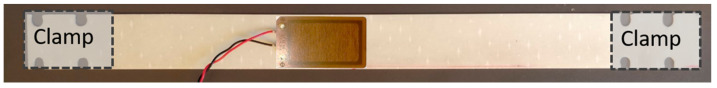
Composite beam of EMCU with bonded piezoelectric transducers.

**Figure 3 sensors-24-02480-f003:**
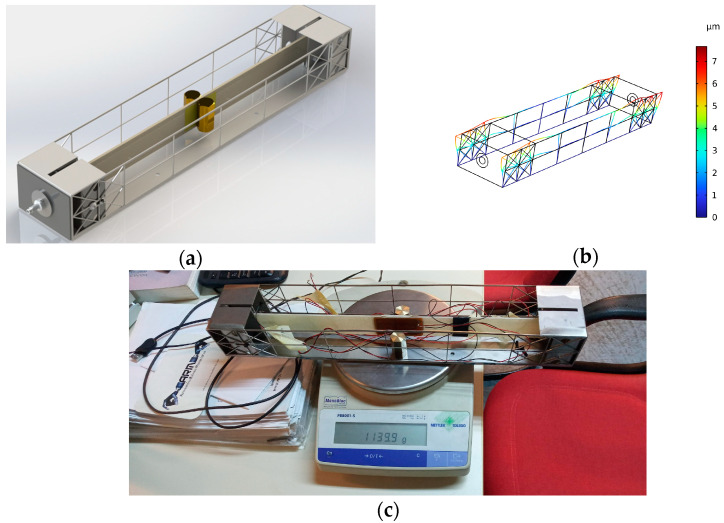
EMCU as designed and manufactured: (**a**) digital mockup, (**b**) displacement of sheet metal frame at axial loads of 400 N and (**c**) physical prototype.

**Figure 4 sensors-24-02480-f004:**
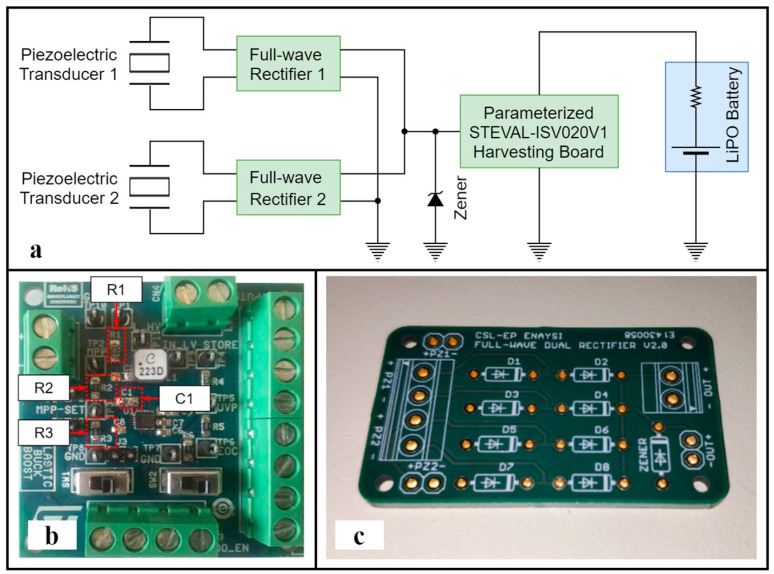
(**a**) Overview of the EHCU architecture. (**b**) Parameterized STEVAL-ISV020V1 harvesting board. Critical components R_1_, R_2_, R_3_ and C_1_ are denoted. (**c**) PCB board integrating two full-wave rectifiers and output overvoltage protection (unsoldered).

**Figure 5 sensors-24-02480-f005:**
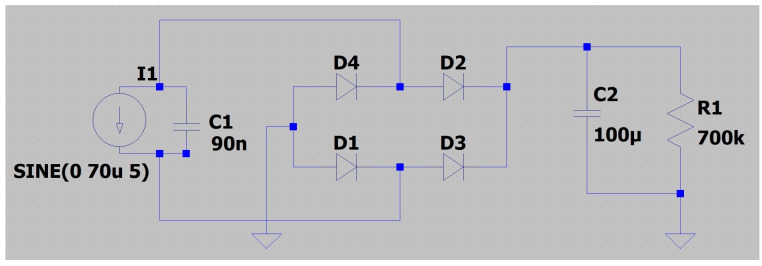
Full-wave rectifier circuit simulated in LTspice XVII.

**Figure 6 sensors-24-02480-f006:**
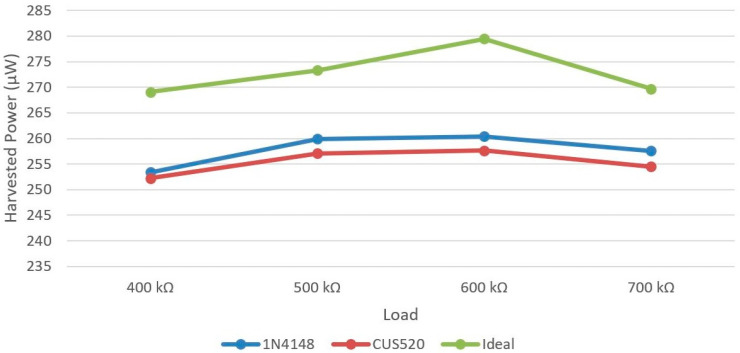
Influence of diode selection for the full-wave rectifier circuit on the mean harvested power.

**Figure 7 sensors-24-02480-f007:**
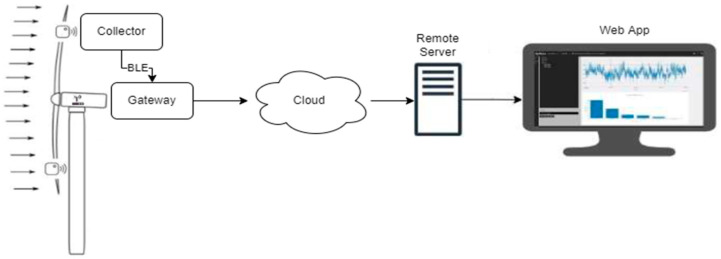
Architecture of the Data Acquisition and Wireless Transmission Unit (DAWTU).

**Figure 8 sensors-24-02480-f008:**
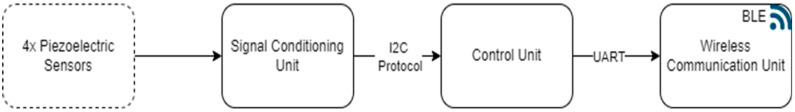
Block diagram format of the architecture of the collector device.

**Figure 9 sensors-24-02480-f009:**
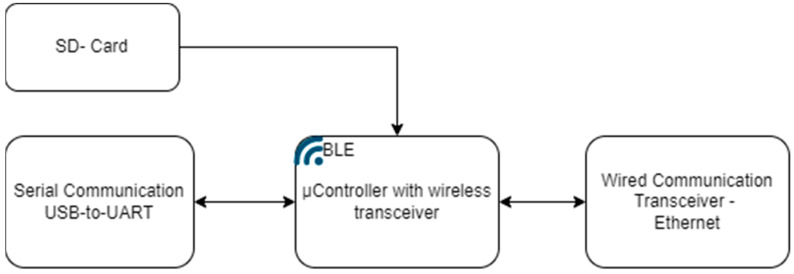
Block diagram format of the architecture of the gateway device.

**Figure 10 sensors-24-02480-f010:**
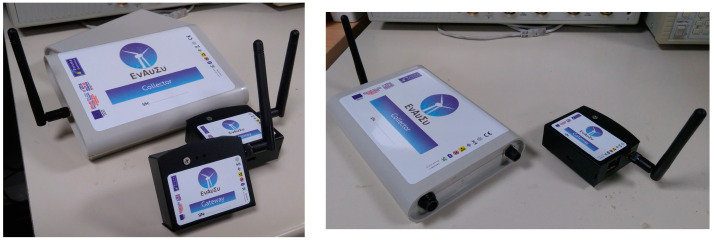
Photos of the physical DAWTU subsystem consisting of a collector and a gateway.

**Figure 11 sensors-24-02480-f011:**
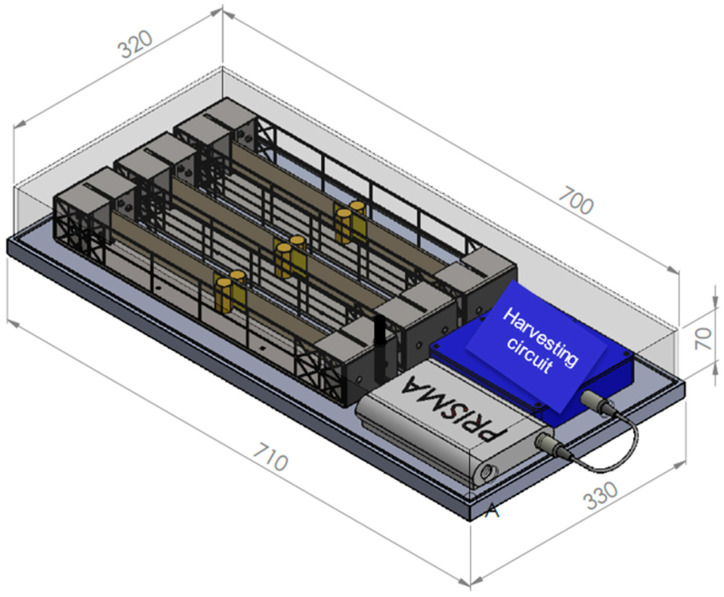
EnAuSy wind turbine blade monitoring system.

**Figure 12 sensors-24-02480-f012:**
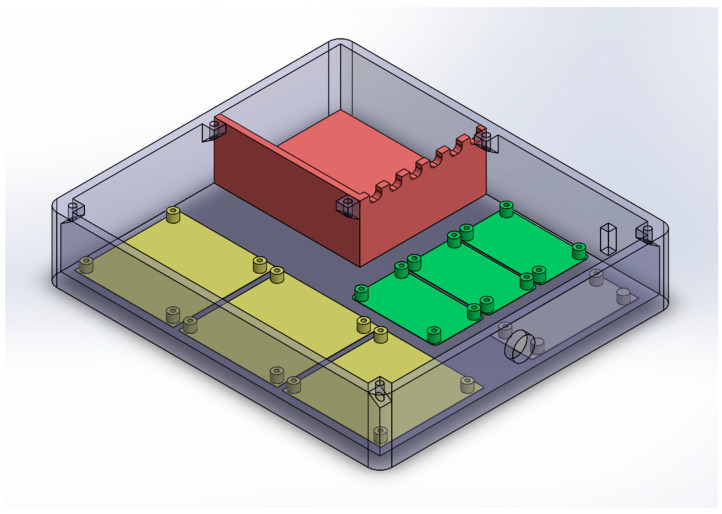
Harvesting circuit subassembly housing design with predefined positions for the battery pack (red), the dual-channel full-wave rectifiers (green) and the harvesting boards (yellow).

**Figure 13 sensors-24-02480-f013:**
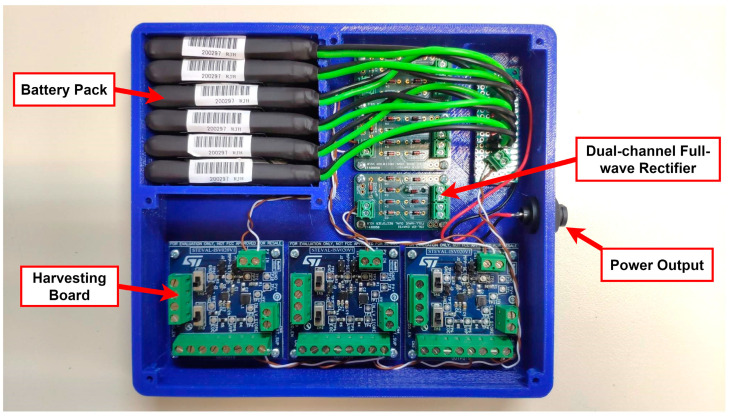
Harvesting circuit subassembly incorporating a battery pack, three dual-channel full-wave rectifiers along with three STEVAL-ISV020V1 harvesting boards and a 2-pin circular power output connector for the DAWTU.

**Figure 14 sensors-24-02480-f014:**
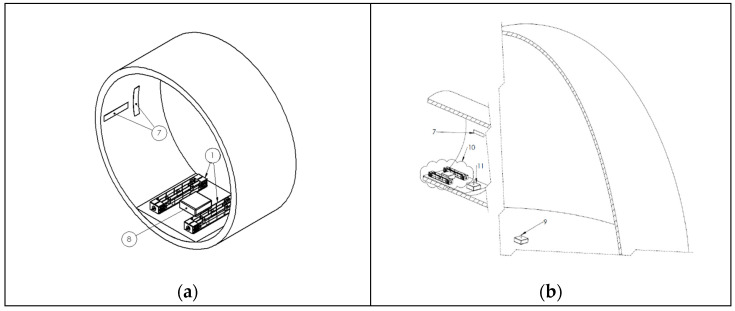
Placement of an EnAuSy Integrated Blade Monitoring System inside the wind turbine blade. The box (**a**) contains a harvesting circuit subassembly and DAWTU except the gateway, which is placed in the nacelle (**b**).

**Figure 15 sensors-24-02480-f015:**
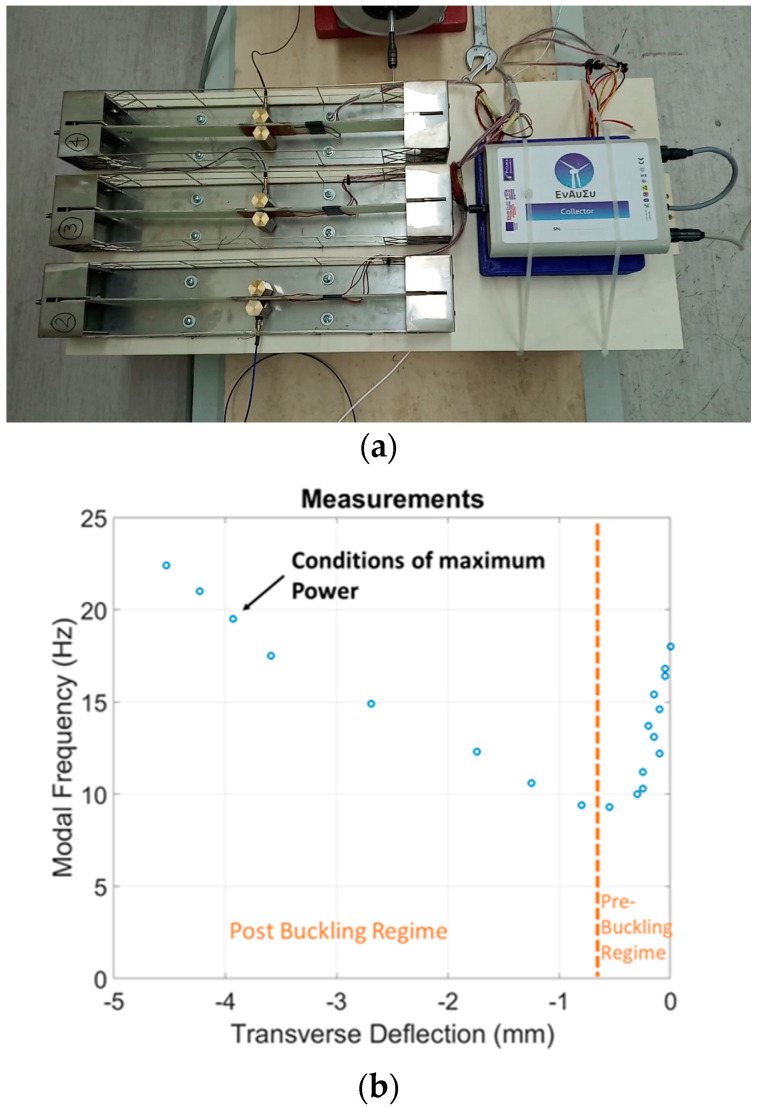
(**a**) Integrated EnAuSy system including three EMCUs, the harvesting circuit subassembly and the DAWTU; (**b**) EMCU calibration, i.e., identification of the most promising beam prestress for maximum power extraction; and (**c**) typical phase plot of bistable beam vibration at post-buckling.

**Figure 16 sensors-24-02480-f016:**
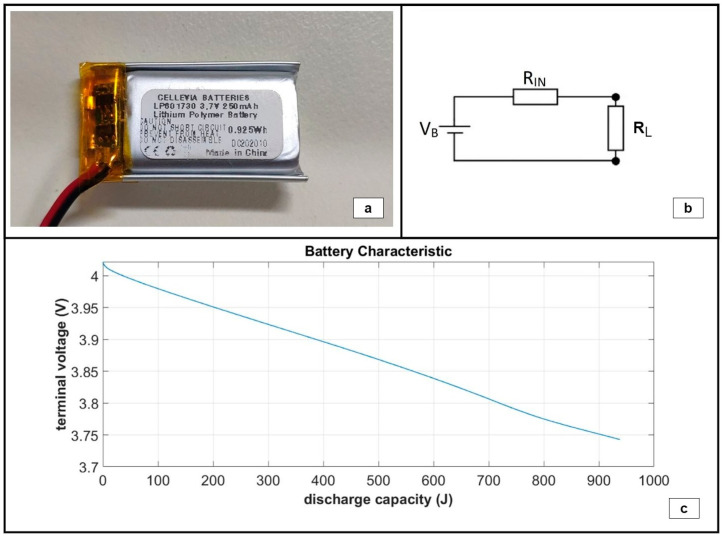
(**a**) Battery used for the ground tests. (**b**) Phenomenological model of the battery under a known constant load, R_L_. (**c**) Resultant characteristic battery curve.

**Figure 17 sensors-24-02480-f017:**
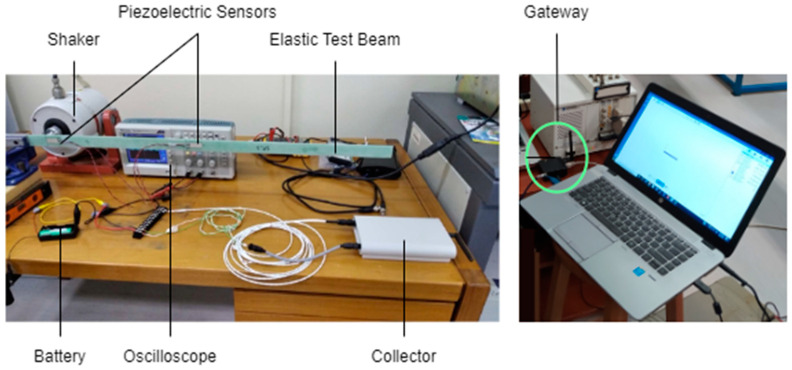
Photos from the experimental setup of the EnAuSy DAWTU subsystem.

**Figure 18 sensors-24-02480-f018:**
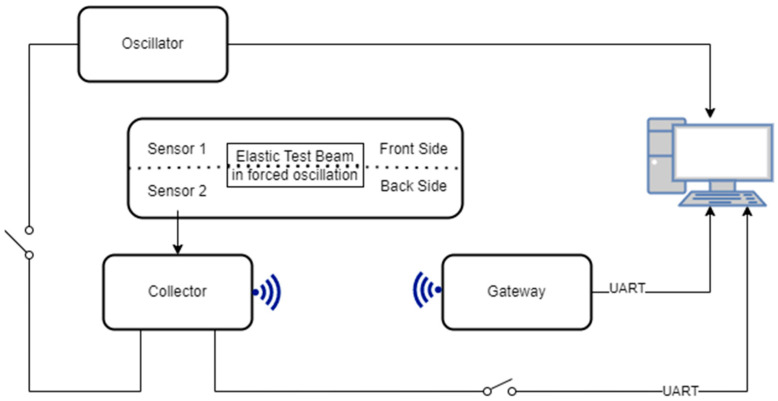
Schematic diagram of the experimental setup of the EnAuSy DAWTU subsystem.

**Figure 19 sensors-24-02480-f019:**
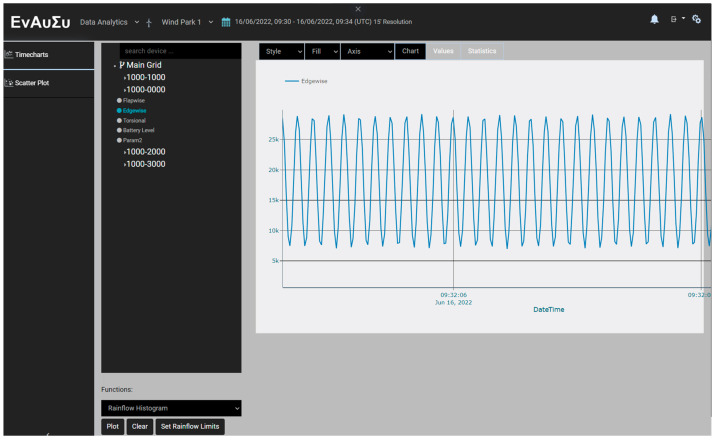
Indicative screenshot from the data display in the web application of EnAuSy.

**Figure 20 sensors-24-02480-f020:**
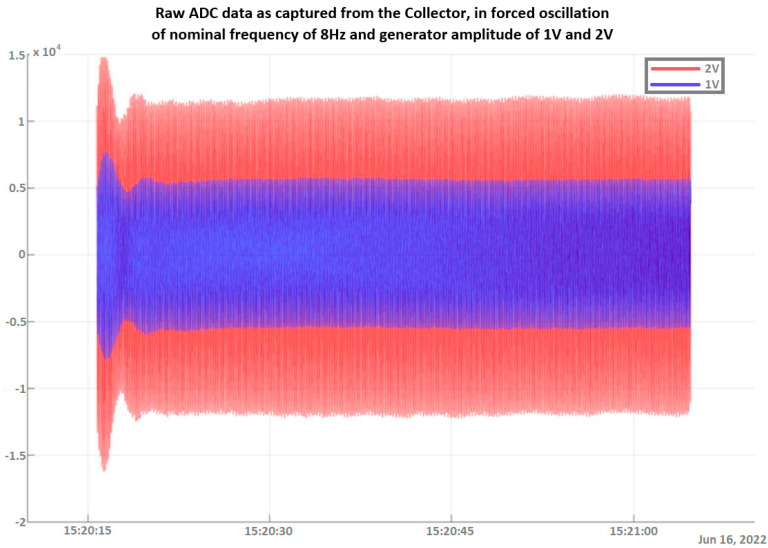
Response signal to an oscillation of a nominal frequency of 8 Hz and a shaker voltage amplitude of 1 V and 2 V as captured via the collector (raw ADC values).

**Figure 21 sensors-24-02480-f021:**
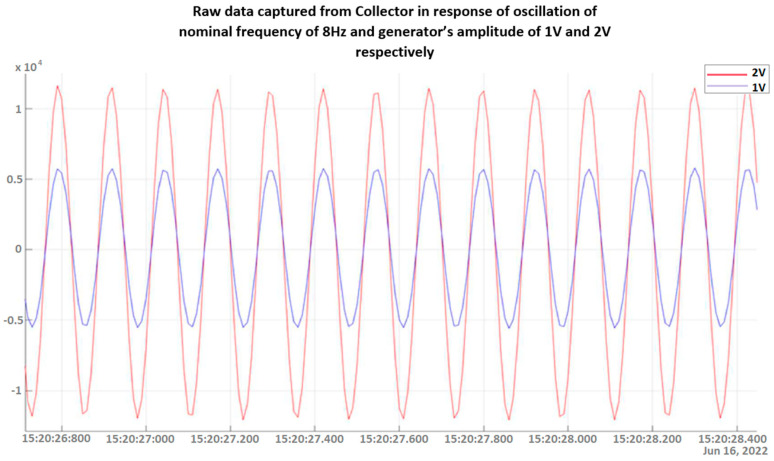
Zooming in on the response signal (raw data) as captured via the collector device to an oscillation of a nominal frequency of 8 Hz and the generator’s amplitude of 1 V and 2 V, respectively.

**Figure 22 sensors-24-02480-f022:**
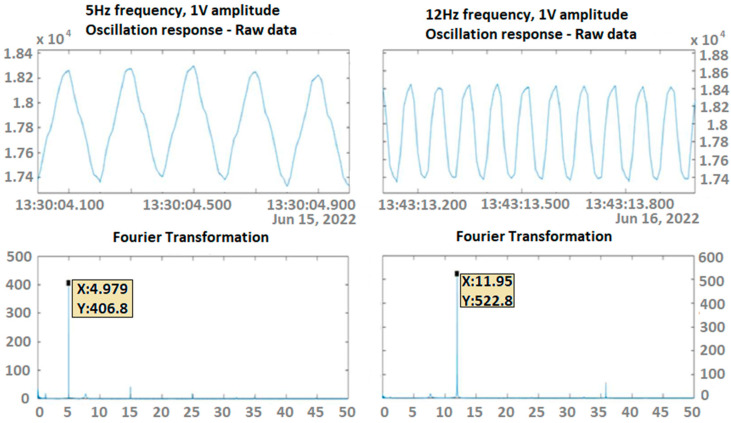
Fourier transformation to determine the main frequency of the recorded signal from the collector when the elastic test beam is subjected to forced harmonic excitation under a 1 V amplitude of shaker voltage at a frequency of 5 Hz (**left**) and 12 Hz (**right**).

**Figure 23 sensors-24-02480-f023:**
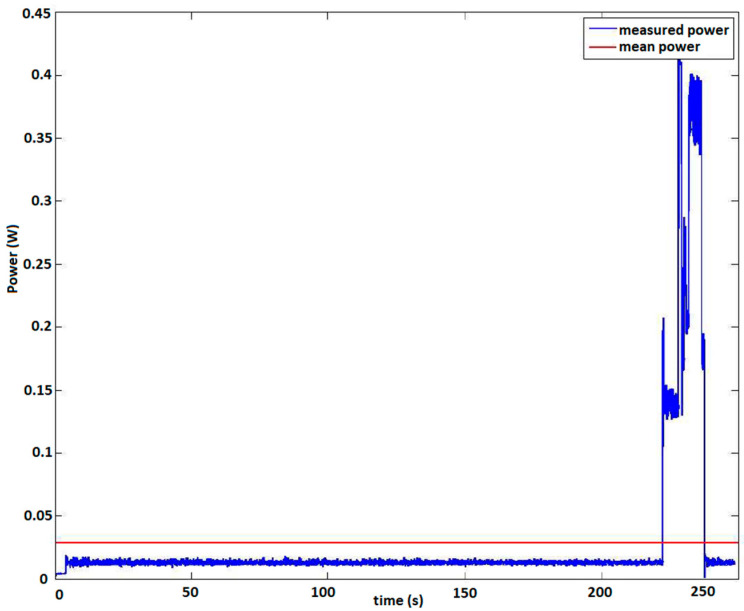
Power consumption of the DAWTU subsystem during one full cycle of operation (from the initiation of the operation until the first wireless transmission of all collected data to the gateway).

**Figure 24 sensors-24-02480-f024:**
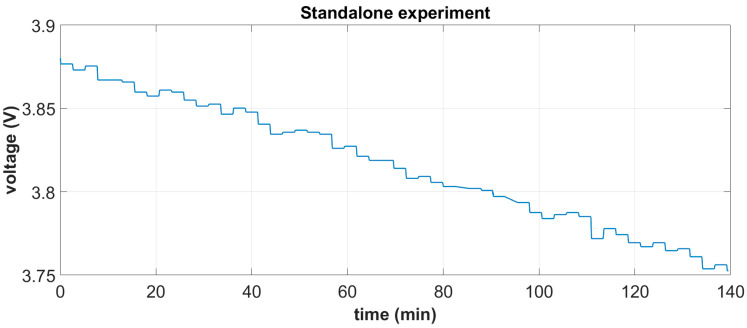
Battery terminal voltage during standalone experiments.

**Figure 25 sensors-24-02480-f025:**
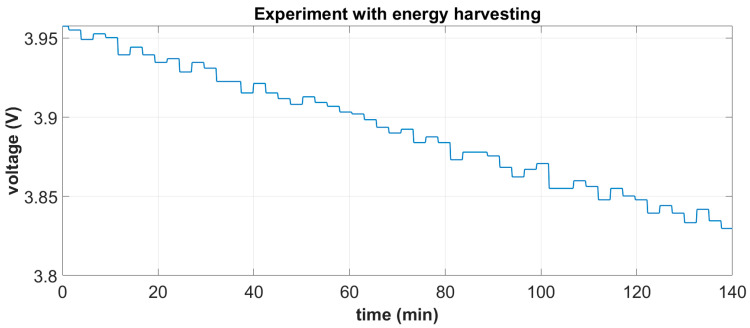
Battery terminal voltage during the experiment with energy harvesting from the electromechanical subsystem.

**Table 1 sensors-24-02480-t001:** Requirements set by the manufacturer.

**Geometrical Requirements**	*Form*	*Dimensional Limits*		
		*Length [mm]*	*Width [mm]*	*Height [mm]*
	Flat	750	350	100
**Operational Requirements**	*Max Weight [kg]*	*Max Frequency of Base Excitation [Hz]*	*Min Power Production [mW]*	*Resilience*
	5	15	1	Yes
	*Sampling* *Frequency*	*Sampled Data Resolution*	*Data Process*	*No. of Sensors per Blade*
	≥10 samples/s	≥13 bit	Rainflow Counting Method	×3 (Capability to Reach 4)
**Other** **Requirements**	*Ease of* *Installation*	*Design* *Flexibility*	*Portability*	*Production Cost*
	Yes	Yes	Yes	Low

**Table 2 sensors-24-02480-t002:** Values of STEVAL-ISV020V1 electric components.

Parameter	Value
R1	10 MΩ
R2	1.2 ΜΩ
R3	1.0 ΜΩ
C1	1 μF

**Table 3 sensors-24-02480-t003:** Comparison of power consumption to determine harvested power.

Configuration Type	Power
Power consumption of DAWTU—electromechanical subsystem connected	46 mW
Power consumption of DAWTU—standalone	53.1 mW
Supplied power from the PEH unit	7.182 mW
Percentage energy profit to the unit	13.52%

## Data Availability

The research data are available upon reasonable request.

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
