# Peer review of "The Design and Ground Test Verification of an Energy-Efficient Wireless System for the Fatigue Monitoring of Wind Turbine Blades Based on Bistable Piezoelectric Energy Harvesting"

_sensors, 2024, doi:10.3390/s24082480_

Round 1

Reviewer 1 Report

Comments and Suggestions for Authors

This long paper described the design, fabrication, integration, and testing of a PEH for the turbine blade monitoring applications.  The need for such a product is obivious, and actual development of such products is highly desirable.  The paper provides complete record of the development and testing, and it can be of great help to product engineers.

As a scientific paper, many important elements are missing.  For instance, the reason for the selection of materials, structures, and performace indicators must be through a systematic analysis with the goal of a better product.  It is clear we do not see a systematic process of analysis with known theory, and it is not the style a scientific paper should be written.  There are journals and magazines on the product development process, and the authors should consider to revise to meet the needs of product engineers.

Due to the lack of theoretical and quantative analyses as a scientific paper, this reviewer would not recommend the publication of this paper.

Reviewer 2 Report

Comments and Suggestions for Authors The author conducted research on energy harvesting and self powered sensing technology for wind turbine blades. In the early stage of related fields, the focus was on the design of collectors, while this article conducted research that is more focused on applications. The specific suggestions are as follows: 1. It is recommended to clarify the power consumption requirements of the system. 2.In order to obtain higher energy in low-frequency rotation, the collector designed in this article has a larger volume. What is the estimated lifespan of the collector, and what are the significant advantages compared to battery replacement? 3. Bistable states require a certain amount of excitation to achieve potential well jumps. Is there a limit on the starting wind speed when the speed of wind turbine blades is usually low?

Reviewer 3 Report

Comments and Suggestions for Authors

The article is devoted to describing and testing of an energy-efficient wireless system for fatigue monitoring of wind turbine blades, where a module with a piezoelectric generator is used as a power source. The article is generally good. The introduction contains enough information to understand the current state of the issue, and the bibliography includes mostly recent work. Methods and approaches are described quite clearly and illustrated. The article contains many illustrations and diagrams that reveal the essence of the work in more detail.

Judging by the figures, the piezoelectric element covers only 20% of the total area of the substrate. Will using a longer piezo element give an increase in power?

Did I understand correctly that the system is supposed to operate in real time?

I was unable to find information in the article about how long it would take to fully charge the battery that was used in the full-scale experiment.

If the piezoelectric generator provides only 23 % of the required power, then is it possible to configure your system so that it transmits data at a certain interval, which would allow the battery to be recharged?

Round 2

Reviewer 1 Report

Comments and Suggestions for Authors

The authors have presented their arguments on the current writing and objective of this paper.  It is not a usual style of scientific papers in engineering due to the lack of innovative procedure and method with details.

The authors have argued that the paper is written in this style is due to the specific advices from the journal and special issue editors.  This is possible, but it is still to be consistent with the usual standard of a techncial journal.

In this case, the decision should be made by the journal editorial office against the journal standard.